# Connections between Asian and European World Cities: Measurement, Analysis, and Evaluation

Ben Derudder [1,2], Xiang Feng [3,*], Wei Shen [4], Rui Shao [2] and Peter J. Taylor [5]

1 Public Governance Institute, KU Leuven, 3000 Leuven, Belgium
2 Department of Geography, Ghent University, 9000 Ghent, Belgium
3 School of Environmental and Geographical Sciences, Shanghai Normal University, Shanghai 200234, China
4 Deakin International, Deakin University, Melbourne 3125, Australia
5 Department of Geography, Northumbria University, Newcastle NE1 8ST, UK
* Correspondence: rachel@shnu.edu.cn

**Abstract:** Although world regions continue to be a key feature of the geographical imagination, there has been relatively little innovative research on world regionalization through the lens of city connections. Against the backdrop of an increasingly urban and interconnected world, in this paper, we evaluate the connections between European and Asian world cities. Based on a model conjecturing intercity connections through office locations of globalized producer services firms, we analyze the networks of both regions' major cities. To this end, we establish frameworks that allow (1) comparison of the level of connectivity of cities and (2) analysis of the strength and orientation of the interactions between cities. We find that both Europe and Asia have a larger number of well-connected cities than any other world region. Both regions are roughly comparable in terms of the distribution of their urban connectivities, but there are some notable differences (e.g., Asia's system being more top-heavy) and evolutions (e.g., Asian cities gaining more connectivity over the last decade). There are also two geographical dimensions to the interpretations of these patterns of urban connectivity: (1) the variegated importance of state-spaces (e.g., national gateways) and (2) the uneven regional focus of intercity connections (e.g., Luxembourg and Singapore being less dependent on regional connections). We use our findings to argue that the time is ripe for a more nuanced and contextualized answer to the question of how cities (can) act politically on the global scale in general and Asia–Europe relations in particular.

**Keywords:** connections; world cities; governance; relations; world regions

## 1. Introduction

Geography has a long tradition of dividing the world into regions for both imperial and pedagogic ends [1]. Such world regions continue to be a key feature of geographical textbooks. However, there has been relatively little innovative research on world regionalization within contemporary globalization [2]. One notable area is the literature on the analysis of cities through the lens of meso-scale regions: there is a rich and evolving body of research analyzing the putatively global connections of cities [3–8] that often finds that these connections are globally ordered [1]. However, these regional orderings are rarely used as the starting point of the analysis. Against this background, the objective of this paper is to examine the strength of the connections between major cities in the world regions that are often described as the erstwhile and likely emerging core of the global economy, respectively: Europe and Asia. A further rationale for focusing on these world regions for a regionally ordered analysis of city connections can be found in the narratives surrounding the Belt and Road Initiative (BRI), which revolves around several priority intercity corridors connecting Asian and European cities. These corridors are epitomized by the land and maritime link maps of the BRI as originally announced by Chinese President Xi Jinping during an official visit to Kazakhstan 2013. This advances an interpretative

geographical framework consisting of interconnected cities between world regions, and this will be the focus of our paper.

This focus on cities and their global connections—now visible in research at different scales [9–15] and in different forms [1,16–20]—in and between both world regions not only serves an empirical, but also a conceptual objective: we aim to rebalance what we believe is a somewhat inflated emphasis on 'the state' as the prime territorial unit of analysis in the literature on the common challenges facing Asia and Europe. Indeed, in spite of 'the urban revolution' [21] sweeping across Asia and Europe, much of the research on these world regions' interactions and common challenges is carried out through the lens of their states rather than through their major cities [22–28]. There are, of course, good reasons for state-centric approaches when studying the interactions between Asia and Europe. As will be shown in this paper, numerous important practices, processes, and institutions guiding the interaction between both world regions continue to be shaped by the context of the now-familiar mosaic of the world geopolitical map. However, there is nothing natural about states being the basic units of analysis in our social-scientific enquiries. Firstly, the world of modern states was made: it is the most obvious example of the more encompassing process of territoriality in which boundaries are used to create clear-cut insides and outsides [29]. As is well known, the modern state was created in two stages. First, territorial sovereignty was consolidated as a process by the Treaties of Westphalia in 1648–9 to impose order and security on Europe. Second, this political mosaic was spread across the world through colonialism. For example, Dick [2005] reminds us that in Asia, contemporary Indonesia emerged out of the colonial state of Java, constructed in the course of the 19th century [30]. The 'Outer Islands' were conquered and subsequently joined onto what would later become the Indonesian nation-state. Maps of contiguous geographic and political space are now taught in school geography and reinforced in the media. In spite of its obvious relevance, this habitual 'way of knowing' has partially locked out other geographical perspectives in public discourse and across the social sciences alike. Our focus on city connections in/between world regions brings these perspectives explicitly to the fore: it can be argued that some aspects of our world are now being unmade and/or increasingly rivaled by alternative geographical imaginaries, such as the one encapsulated by the BRI.

Cities have come to the forefront against the background of the general rise of cross-border economic interactions in an era of liberalized trade and investment [31,32]. For example, from the 1970s onwards multinational corporations have been at the roots of reorganizing the world economy through setting up global production networks in which cities are unevenly embedded. Referring back to the Indonesian example, Indraprahasta and Derudder [2017] recently showed how Indonesia's involvement in the world economy can also be understood through a city network lens: by analyzing Indonesia's major cities' corporate connections cities across the world, they provide evidence of above all Jakarta being strongly embedded in corporate networks that are primarily geared toward East Asian cities [33].

Much research on global-scale city networks draws heavily on the work of Sassen [2018] who defined global cities, such as London, New York, and Tokyo, as centers of advanced services creating new skills, interpretations, and insights within border-crossing networks of knowledge and information flows [3]. This formative research has later been extended in conceptual, analytical, and empirical terms. In this paper, we draw on some of these extensions put forward in the context of the Globalization and World Cities Research Network (GaWC) to evaluate how Asian and European cities are interconnected. The GaWC specification of intercity connections draws on an analysis of the (net)working practices within putatively 'global' service firms. These firms have created city-centered office networks across the world that are reproduced through myriad interactions between their different office locations (e.g., virtual platform telephone, email, online meeting tools, and physical staff travel/exchanges). Using a model that conjectures city connections between these office locations provides the evidential basis which we will use to analyze the networks of and between Asia's and Europe's major cities.

This paper is structured in three main parts. First, the analytical framework is discussed. Second, there is an overview of the main results. Third and finally, we provide a formative interpretation of our results by framing them in the context of possible metanarratives about Asia–Europe connections and reflect on the wider implications of this research.

## 2. Materials and Methods

### 2.1. Theoretical Basis

The makers of cities and city connections are multifarious. Many contemporary world cities clearly reflect their key makers: cultural agency has created Los Angeles as a world media city; political agency has created Geneva as an international institutional city; and economic agency has created Hong Kong as an international financial center. Of course, leading cities are the result of all three of these agencies creating 'well-rounded' world cities, as initially envisaged by Hall [1966] [34]. London and New York are clear-cut examples: both are very important media, political, and financial centers and much else besides—in fact, they simply cannot be ignored by agents pursuing global strategies [35]. In this study, we focus on a specific type of economic agents and their creation of city networks. Following Sassen [2001, 2018], the agents we concentrate upon are producer service providers. Thus, we build upon the traditional urban approach of researching intercity connections through treating cities as service centers: we specify a city network of global service centers [3,36].

However, we do not wish to suggest that service firms act alone in the creation of networks between world cities. The reasons why we privilege the global producer services firms in our approach are twofold. First, following Jacobs [1984], it is the firms as economic agents who produce the wealth upon which the network has been built and is sustained [37]. Second, it is the firms, through their office networks, that have created the overall structure of the network. Since the latter is the focus of this paper, it is to firms that we look to specify and measure the network. Other agencies are not ignored; they play important roles for interpreting network patterns. The most obvious example, and one that will be discussed at some length in the results section, is how states influence the formation of intercity connections. The influence of the state can be theorized to include both economic policies and general cultures of conducting business. For the former, the relative liberalization of national legal and economic policies may be crucial in enabling flows between world cities. A major example is London's 'Big Bang' in 1986, when the British government gave foreign investment banks access to the London Stock Exchange following state-led measures that de-regulated the British financial system. For the latter, a good example is how the culture of saving in Japan has been vitally important for the global growth of Japanese banks.

Nonetheless, we argue that the new global patterns of intercity connections are ultimately the result of the recent rise of large numbers of global service firms, and the empirical basis of our analysis consists of an application of network analysis to data detailing the office locations of global producer services firms across major cities in 2020. The position of this approach in the broader literature on world cities has been summarized in Derudder and Taylor [2016] and Derudder and Taylor [2018]; here, we restrict ourselves to a basic summary to make this paper self-standing [38,39].

Our emphasis on producer services firms is based on Sassen's [2001] observation that major cities derive their status from the (re)production of economic globalization through their development of producer services complexes [36]. These complexes consist of service firms offering tailored financial, professional, and creative expertise to corporate clients. As the latter firms have 'gone global', so did the firms servicing them in areas such as corporate law, management, corporate tax advice, and advertising. The result was that some cities simultaneously became major markets for these services through corporate presences and production centers of these services through interconnected knowledge clusters. These processes have continued across transformations within the producer services sector and emerging global economic geographies, as argued in Sassen's [2019] most recent review of

this conceptual framework. Our analytical approach draws upon Sassen's argument point for analyzing how producer firms connect cities in the global economy [40].

*2.2. Empirical Strategy*

Without official published data, we need to self-organize the collection of information on how producer firms connect cities in the global economy. A potential problem is confidentiality since, as a rule, firms often do not want to reveal their strategies, including locational strategies, to their competitors. However, producer service firms are somewhat different in this crucial respect. For firms that have chosen to pursue a strategy of providing services across the world, their 'global presence' is an integral part of their public marketing and recruitment policies. For instance, new potential clients from around the world will want to know the geographical range of the services on offer. Additionally, since these are knowledge-based firms, a global scope is very obviously an important advantage in signing up the best of the next generation of key professionals. Therefore, among producer service firms, locational strategy is perforce quite transparent. Figure 1 presents straightforward examples of this. The pictures in Figure 1 were taken at Amsterdam's Schiphol Airport in April 2004. They were part of a marketing campaign of what was then Deloitte. The campaign's message centers on Deloitte's global presence: pictures of iconic buildings in self-evident world cities—New York's Chrysler Building and Amsterdam's crow-stepped gables—are used to suggest that Deloitte is located in all cities that 'matter' (at least in terms of where Deloitte can do profitable business). Put simply: the message for prospective clients is that there can be no doubt that Deloitte has an office in the likes of Paris, Chicago, Tokyo, and São Paulo. If in doubt, clients can turn to the corporate websites of such firms. Indeed, another integral part of the showcasing of the geographical range of the services on offer is that these websites provide an option to select 'location', giving the addresses of their offices. The map shown in Figure 2 is an example of this. It shows a world map with the geographical distribution of the Deloitte offices to emphasis their global presence. Not all major producer services firms have a geographical reach as extensive as Deloitte's, but the idea is that advantage is taken of this geographical transparency for information gathering.

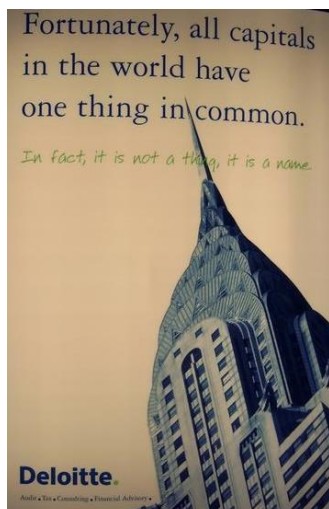 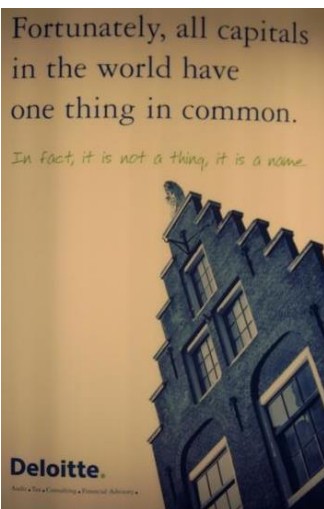

**Figure 1.** Marketing campaign of Deloitte at Amsterdam's Schiphol Airport in April 2004. Source: Taylor and Derudder (2015).

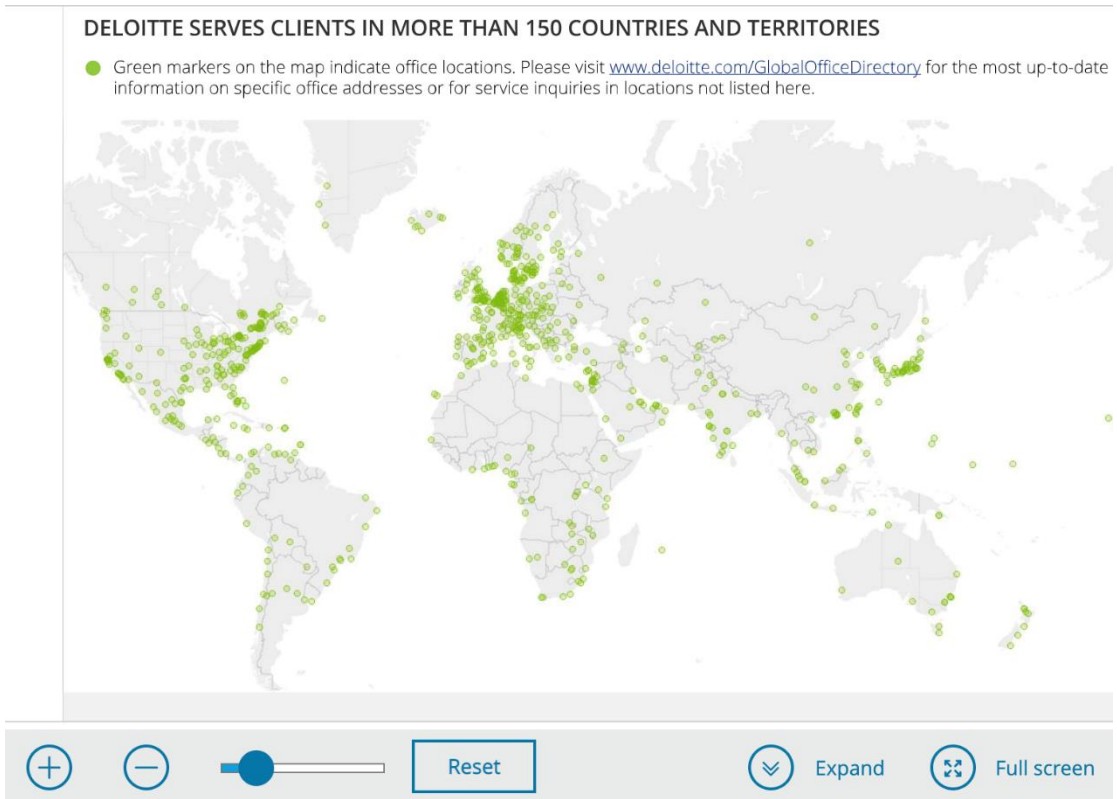

**Figure 2.** Website of Deloitte providing an option to select 'location', giving addresses of offices. Source: https://www.deloitte.com/GlobalOfficeDirectory, accessed on 20 August 2022.

Firms are selected based on rankings of leading service firms in key producer services sectors, a straightforward approach that has the advantage of facilitating future replication of the data gathering. In recent data gatherings, and also the one underlying this paper, we focused on the leading firms in 5 sectors: 75 financial services firms, 25 management consultancy firms, 25 advertising firms, 25 law firms, and 25 accounting firms. The information on the location strategies of these firms was gathered from July–August 2020. Firms were selected based on sectoral rankings for 2019, which tended to be based upon 2018 data. We selected financial services firms from BrandFinance's Top 500 financial services and insurance companies, which is based on a benchmark study of the strength, risk, and future potential of financial services firms; accounting firms were chosen from World Accounting Intelligence's ranking, which is based on an analysis of aggregated company revenues; advertising agencies were selected based on Brandirectory's analysis of the valuable brands in the advertising sector; law firms were selected based on Chambers' ranking of leading corporate law firms; and management consultancy firms were selected from the Vault Management and Strategy Consulting Survey, which ranks firms in terms of their 'prestige' based on a large survey of professionals. For each sector, the top-ranked firms were chosen, and we also identified substitute firms (i.e., ranked just below 75 and 25) to cover for situations in which a firm disappeared (e.g., been taken over) during the actual data collection. Note that although our starting point is firms, the information we collected defines firm networks with very different levels of corporate integration [Jones 2002] [41]. Alongside tightly organized global firms operating under a single corporate flag (e.g., PricewaterhouseCoopers), there are also 'firms' that are in fact groups of firms (e.g., Leading Edge Alliance Group). In the latter case, the firm is in fact an alliance of medium-sized firms constituted as a network to compete globally with the very large firms leading this sector. Appendix A gives an overview of the final list of firms.

A few of the larger firms have branches in many hundreds, even thousands, of cities and towns (cf. Figure 2). However, data collection has been restricted to the more important

cities for two reasons. The first is analytical: the more cities included, the sparser the final matrix will become, with nearly none of 175 firm networks present in the smaller cities and towns. The second is theoretical: the interest is in the more important intercity connections. Nevertheless, it is also important not to omit any possible significant node in the world city network so that a relatively large number of cities need to be selected. Our selection is based on a number of overlapping criteria. In addition to the original 315 cities that featured in Taylor [2004], we also included all cities with a population of more than 1.5 million inhabitants in 2018 [42]; all capital cities of states with a population of more than one million, and every city with a headquarter office of one of our selected firms. This led to the selection of 707 cities listed in Appendix B.

The final step is gathering information on the importance of a given city to a firm's global service provision. There is no simple, consistent set of information available across firms. The prime sources of information are websites, and these differ greatly among the 175 firms. It is necessary to scavenge all possible relevant available information, firm by firm, from these sites supplemented by material from any other sources available such as annual reports and internal directories. For each firm, two types of information have been gathered. First, information about the size of a firm's presence in a city is obtained. Ideally, information on the number of professional practitioners listed as working in the firm's office in a given city is needed. Such information is widely available for law firms but is relatively uncommon in other sectors. Here, other information must be used, such as the number of offices the firm has in a city. Second, the extra-locational functions of a firm's office in a city are recorded. Headquarter functions are the obvious example, but other features, such as subsidiary HQs and regional offices, are recorded. Any information that informs these two features of a firm's presence in a city is collected in this scavenger method of information gathering. This standardization involved assigning service values $v_{ij}$ for firm $j$ in city $i$, with $v_{ij}$ ranging from 0 to 5 based on the following procedure. The city housing a firm's global headquarters was scored 5; a city with no office of that firm was scored 0. An 'ordinary' or 'typical' office of the firm resulted in a city scoring 2. With something missing (e.g., no partners in a law office), the score reduced to 1. Particularly large offices were scored 3, and those with important extra-territorial functions, such as regional headquarters, were scored 4. The end result was a 707 × 175 service value matrix *V* in which *vij* ranges from 0 to 5.

### 2.3. Specification of Basic Connections between Cities

Broadly, there are two ways in which the data contained in the services value matrix V can be analyzed. The first approach is to directly examine the matrix using multivariate analysis [43]. The second approach is to transform the information contained in the matrix into connectivity measures specifying the relative strength of connections between cities [44]. Given that our interest is in city connections, we follow this second approach. In the literature, different types of transformations from the service value matrix to a connectivity matrix have been put forward [45]. Here, we use a bespoke transformation function: the interlocking network model (INM) initially put forward in Taylor [2014] to analyze producer services firms' networks [14].

The cornerstone of the INM is the formal specification of city-dyad connectivity $CDC_{a-i}$ between cities a and i, defined as follows:

$$CDC_{a-i} = \sum v_{ai} \cdot v_{ij} \tag{1}$$

where $CDC_{a-i}$ details the potential knowledge and information flows between a pair of cites a and i based upon the assumption that the more important an office, the more intercity working connections it produces. It can be said that the INM resembles a straightforward interaction model. This transformation builds a tentative answer to, for example, the following issue: if someone entered the Shanghai office of a major Chinese bank, what level of service could they expect for their business needs in other cities? One would expect major service for dealings in London since most key Chinese banks now have a

presence in London. However, what if service is needed for business in Amsterdam or Dresden? Certainly, the probabilities of the banks having an office in these cities will be smaller than for London, and if they are present, then the degree of service offered would likely be less than in London. The INM provides a method of estimating these levels of service connections across firms. Of course, globalized producer service firms' networks diverge in their geographical coverage and how they operate their network: they are specific, depending on a firm's geographical foundations, its client basis, and its internationalization strategy and business model. As a consequence, this approach to formally specifying intercity connections depends upon aggregating all 175 office networks to deal with these specifics.

Equation (1) provides the pillar of the GaWC method towards measuring city connections. It has been used and extended in various ways to answer different research questions. In this paper, we draw on some of these extensions to establish a framework for comparing and relating cities in Asia and Europe. In previous research [38], we organized the 707 cities into 9 world regions that commonly return in transnational organizational schemes: Africa/Middle East, West/Southwest Asia, North America, Central and South America, Europe, and Pacific Asia/Oceania. For the purpose of this paper, we reorganized the geographical allocation so that 'Europe' equals Europe and the parts of Eurasia roughly west of the Caspian Sea and the Ural Mountains, while 'Asia' equals Pacific Asia, South Asia, and the parts of Eurasia east of the Caspian Sea and the Ural Mountains. This geographical division is—like any other alternative—of course not without its flaws and to some degree anathema to the notion of city networks providing an alternative to areal classifications, but it is used here as a heuristic that allows us to broadly capture some of the parallels and differences between both world regions.

The remainder of the presentation of our analytical framework proceed in two steps: first, we establish a framework that allows comparing the level of connectivity of cities in Asia and Europe; second, we establish a framework that allows the analysis of the strength and orientation of the interactions between cities in Asia and Europe. Readers are referred to Derudder and Taylor [2016] and Derudder and Taylor [2018] for more background, details, and descriptions; here, we focus on straightforward and non-technical descriptions of the different measures as well as their intuitive interpretation [38,39].

### 2.4. Comparing Cities in Asia and Europe

To compare the evolving position of Asian and European cities in the global networks of producer services firms, we formulate two measures: global network connectivity and change in global network connectivity. First, a city's global network connectivity ($GNC_a$) can be calculated by simply aggregating all its connections as per Equation (1):

$$GNC_a = \sum \sum_{ij} v_{ai} \cdot v_{ij} \qquad (2)$$

To make the GNC measures in Equation (2) independent from the number of cities/firms, we report connectivities as percentages of the most connected connectivity in the data, thus creating a scale from 0% (no connections) to 100% (London). In the remainder of our paper, we will focus on the 37 Asian and 39 European cities that have a $GNC_a > 25\%$.

Second, cities' global network connectivities in 2020 provide a recent snapshot, the present-day outcome of variegated trajectories in the past. To be able to assess cities' connectivity trajectories, we draw on an earlier post-global financial crisis data gathering in 2010 to measure the evolution of cities' $GNC_a$ between 2010 and 2020. This is done through measures of connectivity change ($CC_a$), detailed in Derudder and Taylor (2016) [38]. These $CC_a$ measures are conceived in such a way that they can be interpreted as a z-score: cities with a CC value > 2 have witnessed exceptional connectivity gains between 2010 and 2020, while cities with a CC value < −2 have witnessed exceptional connectivity losses between 2010 and 2020. Cities with a value close to 0 have retained the same level of connectivity between 2010 and 2020.

*2.5. Connecting Cities in Asia and Europe*

To assess the strength of the connections between Asian and European cities in the global networks of producer services firms, we formulate two measures: relative city-dyad connectivity and regional orientation.

First, to examine how strongly cities in Europe and Asia are connected, we develop an alternative measure that allows for a more refined appraisal of intercity connections than the one given by Equation (1). These measures of relative city-dyad connectivity ($RCDC_{a-i}$) put connectivities in the context of both cities' level of $GNC_a$. This alternative measure is calculated as follows:

$$RCDC_{a-i} = CDC_{a-i}/(GNC_a \times GNC_i) \tag{3}$$

Again, to make $RCDC_{a-i}$ manageable (i.e., independent from the number of cities/firms), we express connectivities as proportions of the most connected city in the data, thus creating a scale from 0% to 100%.

Second, each city has connections with cities in the own world region (e.g., Singapore with Hong Kong, or Paris with Frankfurt) and with cities in the other world region (e.g., Singapore with Paris, or Frankfurt with Hong Kong). To analyze the dominant geographical orientation of each city's connections, we divide its aggregated $RCDC_{a-i}$ values with cities within the own region by its aggregated $RCDC_{a-i}$ values with cities in the other region to produce a measure of regional orientation ($RO_a$) detailed in Taylor and Derudder [2015] [46]; values > 1 point to the connections within the own region being stronger than connections with the other region, values = 1 point to connections within and outside of the region being in balance, and values < 1 point to connections outside the own region being stronger than connections within the own region. The farther away from the midpoint of 1, the stronger the tendency of that regional orientation. Table 1 provides an overview of the four measures that will be used in the next section to examine the global connections of Asian and European cities.

**Table 1.** Description and interpretation of city connectivity measures.

| | Measure | Description | Interpretation of Reported Measure |
|---|---|---|---|
| Asia–Europe Comparison | Global Network Connectivity (GNC) | Aggregated connectivity across the global economy | % of GNC of most connected city:<br>• 100% for most connected city<br>• 0% for unconnected city<br>Standard deviation: |
| | Connectivity Change (CC) | Shifting level of GNC in the periods 2000–2020 and 2010–2020 | • >2 for major connectivity gains<br>• <−2 for major connectivity losses<br>• values between −1 and +1 point to relative stability |
| Asia–Europe Connectivity | Relative City-Dyad Connectivity (RCDC) | Strength of intercity connectivities | % of RCDC of most connected city-pair:<br>• 100% for most connected city-pair<br>• 0% for unconnected city-pair |
| | Regional Orientation (RO) | Relative balance between relative city-dyad connectivities inside and outside the own region | • >1: connections inside the own region stronger than with the other region<br>• =1: connections inside and outside the region in balance<br>• <1: connections outside the own region stronger than within the own region |

## 3. Results

*3.1. Comparing Cities in Asia and Europe*

Table 2 gives an overview of all cities in Asia and Europe with a $GNC_a$ of at least 25%. Europe and Asia have a larger number of cities with this level of connectivity than any other world region: 39 cities in Europe and 37 cities in Asia exceed this threshold.

There is a broad range of levels of connectivity amongst both world regions' cities, ranging from London and Hong Kong to the likes of Saint Petersburg and Pune. Put differently, in both Asia and Europe, there is a combination of cities of variable importance. Nonetheless, at the top of the ranking, there is a disjuncture in that by far the most connected city is located in Europe on the one hand, while there are more well-connected cities in Asia on the other hand; although Hong Kong—the third-most connected city globally after London and New York–does not quite have London's prowess, there are five Asian cities (Hong Kong, Singapore, Shanghai, Beijing, and Tokyo) with a $GNC_a > 60\%$ versus only two European cities exceeding that threshold (London and Paris). In a 2008 cover story in Time Magazine, it was argued that Hong Kong would join London and New York, leading to 'three connected cities driving the global economy', but our analysis suggests that rather than Hong Kong performing that specific role, there seems to be a more polycentric pattern of well-connected cities in Asia, leading to a relatively top-heavy pattern.

**Table 2.** An overview of all cities in Asia and Europe with a GNCa of at least 25%.

| Rank | City | GNC | Rank | City | GNC |
|------|------|-----|------|------|-----|
| 1 | London | 100.00 | 1 | Hong Kong | 70.32 |
| 2 | Paris | 60.57 | 2 | Singapore | 65.36 |
| 3 | Amsterdam | 53.94 | 3 | Shanghai | 64.72 |
| 4 | Milan | 52.75 | 4 | Beijing | 64.10 |
| 5 | Frankfurt | 52.24 | 5 | Tokyo | 60.35 |
| 6 | Madrid | 48.77 | 6 | Mumbai | 54.22 |
| 7 | Moscow | 48.70 | 7 | Kuala Lumpur | 49.12 |
| 8 | Brussels | 47.90 | 8 | Jakarta | 48.48 |
| 9 | Warsaw | 46.31 | 9 | Seoul | 46.16 |
| 10 | Zurich | 45.07 | 10 | Bangkok | 44.27 |
| 11 | Stockholm | 43.57 | 11 | Guangzhou | 43.44 |
| 12 | Vienna | 43.56 | 12 | Taipei | 42.50 |
| 13 | Dublin | 43.36 | 13 | New Delhi | 41.28 |
| 14 | Munich | 42.06 | 14 | Manila | 40.59 |
| 15 | Luxembourg | 42.02 | 15 | Shenzhen | 40.40 |
| 16 | Prague | 39.53 | 16 | Bangalore | 40.12 |
| 17 | Lisbon | 39.44 | 17 | Chengdu | 35.94 |
| 18 | Hamburg | 38.36 | 18 | Ho Chi Minh City | 33.77 |
| 19 | Rome | 37.78 | 19 | Tianjin | 33.21 |
| 20 | Berlin | 37.66 | 20 | Chennai | 30.70 |
| 21 | Barcelona | 35.71 | 21 | Nanjing | 30.61 |
| 22 | Düsseldorf | 35.69 | 22 | Hanoi | 30.45 |
| 23 | Bucharest | 35.14 | 23 | Hangzhou | 30.06 |
| 24 | Budapest | 35.10 | 24 | Karachi | 29.52 |
| 25 | Copenhagen | 34.74 | 25 | Chongqing | 29.34 |
| 26 | Athens | 33.72 | 26 | Wuhan | 29.25 |
| 27 | Kiev | 32.09 | 27 | Osaka | 29.04 |
| 28 | Helsinki | 31.26 | 28 | Changsha | 28.16 |
| 29 | Oslo | 31.15 | 29 | Zhengzhou | 27.22 |
| 30 | Geneva | 29.15 | 31 | Xiamen | 27.14 |
| 31 | Manchester | 29.08 | 30 | Dhaka | 27.14 |
| 32 | Stuttgart | 28.78 | 32 | Shenyang | 27.02 |
| 33 | Belgrade | 28.64 | 33 | Almaty | 26.92 |
| 34 | Sofia | 27.79 | 34 | Xi'an | 26.21 |
| 35 | Bratislava | 27.75 | 35 | Dalian | 26.09 |
| 36 | Zagreb | 27.30 | 36 | Jinan | 25.43 |
| 37 | Lyon | 26.85 | 37 | Pune | 25.12 |
| 38 | Nicosia | 26.42 | | | |
| 39 | St Petersburg | 25.96 | | | |

Even though our research deals with a metageographical model of a global economy consisting of interconnected urban economies, this obviously does imply that markets

operate at just these two scales. The idea of the 'national economy' as a closed system may be a myth, but there are nonetheless national market implications for cities as globally interconnected service centers. Jacobs [1984] posited national urban development processes that favor a specific city over all others in a country. Such a process provides that city with a particularly strong platform on which to globalize, especially as new firms begin a global strategy and plan to serve national markets through just a single office [37]. The relevance of this national gateway function, which highlights the continued bearing of the notion of 'urban primacy' [47], is often assumed by the capital city. This phenomenon also explains why many of the European cities in Table 2 have connectivities that are on par with Asian cities even though they are on average smaller.

The most basic empirical manifestation of this capital city gateway pattern is that in both Asia and Europe, we often find that a country's capital is the only well-connected city in Table 2. Furthermore, this city's global network connectivity often broadly reflects the size of the national economy: in Asia, this is visible in Indonesia (Jakarta), Malaysia (Kuala Lumpur), the Philippines (Manila), Bangladesh (Dhaka), South Korea (Seoul), and Thailand (Bangkok); in Europe, this pattern can be found in inter alia Belgium (Brussels), Austria (Vienna), Sweden (Stockholm), Norway (Oslo), Denmark (Copenhagen), Finland (Helsinki), Poland (Warsaw), Ireland (Dublin), Czechia (Prague), Hungary (Budapest), Slovakia (Bratislava), Bulgaria (Sofia), Croatia (Zagreb), Serbia (Belgrade), Portugal (Lisbon), Greece (Athens), and Ukraine (Kiev). Furthermore, even when a second city features in Table 2, the differences in global network connectivity can be quite stark. For example, even though Osaka and Lyon also appear in Table 2, it is clear that Tokyo and Paris are—in relative terms—much more globally connected than might be expected based on the size of their respective urban economies alone. Hill and Fujita [1995] have referred to this situation as 'Osaka's Tokyo problem', but Table 2 suggests that this is a shared experience across Asia and Europe rather than a Japanese phenomenon [48].

There are a number of intersecting and overlapping exceptions to this basic capital city gateway pattern. A first exception is that sometimes, non-capital cities that are economic or financial hubs, such as Mumbai, Milan, and Frankfurt, are more connected than their capital cities (Delhi, Rome, and Berlin, respectively). Second, some cities clearly perform a role that transcends the national market, with Singapore, London, and Luxembourg as obvious examples. Third, in states with notably low levels of political centralization and/or long-standing economic rivalries between (the elites of) its major cities, there are often two or more cities with sizable global connectivities—for example, in Spain (Barcelona and Madrid) and India (Mumbai and Delhi). At its most extreme, there are polycentric urban systems, with different cities having fair levels of global network connectivity. This is the case in Germany (with Frankfurt, Munich Berlin, Hamburg, Düsseldorf and, Stuttgart) and China (with a 'tri-primate' pattern centered on Hong Kong–Shanghai–Beijing alongside a long tail of well-connected cities). The German and Chinese examples also reveal that larger economies require more than one world city to service sub-national regions and ensure a coordinated development trajectory. Fourth and finally, there are a number of miscellaneous patterns—states where (historical) political divisions inhibited the emergence of a single leading city (e.g., Hanoi and Ho Chi Minh City in Vietnam) and states that have developed a functional division of labor between a capital city and one or more commercial gateways (e.g., Milan and Rome in Italy, Karachi and Rawalpindi/Islamabad in Pakistan). This then develops in two or more cities with sizable connectivity, as producer services firms need to be both where political and economic decision-making take place.

The above processes converge in China, where Beijing and Shanghai—alongside the 'special administrative region' of Hong Kong—tower over the other major Chinese cities. This is the joint result of the extent of the national market making it difficult to work from one city; its political divisions, with Hong Kong still operating as a quasi-autonomous area in financial and economic terms [49]; hegemonic ideas of inter-city competition among some of China's urban elites—for example, in Shanghai [50]; and functional divisions of labor as the Chinese political system imposes a context in which producer services

must be near the core of political decision-making in Beijing irrespective of commercial opportunities in Shanghai or Hong Kong [51].

The presence of Hong Kong, Shanghai and Beijing among the world's most connected cities echoes the rise and growing global entanglement of China more generally [1,43,51–55]. This leads us to Table 3, which shows the same set of cities rank-ordered by their $CC_a$ between 2010 and 2020. While the pattern summarized in Table 2 suggests that today, both world regions have roughly similar levels of connectivity, it is clear that this present-day snapshot is the result of uneven trajectories, with European cities having retained similar levels of global connectivity and Asian cities having gained global connectivity. When calculating the average level of $CC_a$ for both word-regions, we find stability for Europe (0,00) and major connectivity gains for Asia (1,20). It has often been argued that the global economy is undergoing a major geographical shift from 'West' to 'East' [55,56], and this is also clearly visible in terms of changes in urban connectivity. Although these rising levels of connectivity are found across Asia, it is above all Chinese cities that stand out; 9 out of 10 Asian cities with the largest connectivity gains are located in China. These are often second-tier cities, suggesting a trend in which after the major gains for the leading cities in the period 2000–2010 [38], connectivity is now spreading to other Chinese cities.

**Table 3.** Cities rank-ordered by their CCa between 2010 and 2020.

| Rank | City | Standardized Connectivity Change 10–20 | Rank | City | Standardized Connectivity Change 10–20 |
|---|---|---|---|---|---|
| 1 | London | 2.04 | 1 | Chengdu | 3.66 |
| 2 | Stockholm | 1.25 | 2 | Changsha | 3.40 |
| 3 | Luxembourg | 1.18 | 3 | Zhengzhou | 3.21 |
| 4 | Amsterdam | 0.98 | 4 | Wuhan | 3.04 |
| 5 | Warsaw | 0.95 | 5 | Chongqing | 2.91 |
| 6 | Belgrade | 0.79 | 6 | Jinan | 2.78 |
| 7 | Helsinki | 0.68 | 7 | Shenyang | 2.71 |
| 8 | Bucharest | 0.50 | 8 | Xiamen | 2.68 |
| 9 | Zagreb | 0.44 | 9 | Hangzhou | 2.40 |
| 10 | Lyon | 0.43 | 10 | Dhaka | 2.37 |
| 11 | Zurich | 0.34 | 11 | Nanjing | 2.29 |
| 12 | Vienna | 0.33 | 12 | Tianjin | 2.23 |
| 13 | St Petersburg | 0.23 | 13 | Xi'an | 2.18 |
| 14 | Hamburg | 0.22 | 14 | Shenzhen | 2.17 |
| 15 | Prague | 0.21 | 15 | Dalian | 1.54 |
| 16 | Berlin | 0.13 | 16 | Guangzhou | 1.22 |
| 17 | Frankfurt | 0.07 | 17 | Beijing | 0.96 |
| 18 | Brussels | 0.06 | 18 | Bangalore | 0.93 |
| 19 | Stuttgart | 0.01 | 19 | Hanoi | 0.78 |
| 20 | Budapest | −0.06 | 20 | Manila | 0.69 |
| 21 | Rome | −0.08 | 21 | Bangkok | 0.68 |
| 22 | Geneva | −0.13 | 22 | Pune | 0.59 |
| 23 | Munich | −0.16 | 23 | Shanghai | 0.57 |
| 24 | Sofia | −0.17 | 24 | Mumbai | 0.32 |
| 25 | Dublin | −0.17 | 25 | Osaka | 0.31 |
| 26 | Lisbon | −0.21 | 26 | Almaty | 0.31 |
| 27 | Bratislava | −0.28 | 27 | Taipei | 0.18 |
| 28 | Kiev | −0.41 | 28 | Jakarta | 0.16 |
| 29 | Copenhagen | −0.47 | 29 | Kuala Lumpur | 0.10 |
| 30 | Milan | −0.64 | 30 | Singapore | 0.05 |

**Table 3.** *Cont.*

| Rank | City | Standardized Connectivity Change 10–20 | Rank | City | Standardized Connectivity Change 10–20 |
|------|------|------|------|------|------|
| 31 | Oslo | −0.70 | 31 | Tokyo | −0.07 |
| 32 | Düsseldorf | −0.75 | 32 | Hong Kong | −0.12 |
| 33 | Athens | −0.75 | 33 | Ho Chi Minh City | −0.44 |
| 34 | Manchester | −0.79 | 34 | New Delhi | −0.47 |
| 35 | Madrid | −0.82 | 35 | Seoul | −0.64 |
| 36 | Moscow | −0.83 | 36 | Chennai | −0.68 |
| 37 | Paris | −0.92 | 37 | Karachi | −0.76 |
| 38 | Nicosia | −1.02 | | | |
| 39 | Barcelona | −1.41 | | | |
| AVERAGE | | 0.00 | | | 1.20 |

Although the differences between Asia and Europe are clear, they are far from homogenous. In Asia, we see slightly declining levels of connectivity in Karachi and Seoul, while—with the exception of Beijing and Shanghai—its major cities have broadly retained the same level of connectivity over the last decade. In the case of Tokyo, a stagnating Japanese economy is clearly responsible for this decline. In the case of Taipei, it seems reasonable to assume that Mainland China's 'opening-up' and the concomitant rising levels of connectivity of its cities has had an effect. Meanwhile, in Europe, there has been another type of west-to-east shift, with nearly all Eastern European cities having gained connectivity between 2010 and 2020 and all above cities in Western Europe having lost connectivity. Cities qualifying as international financial centers, such as London and Luxembourg [57], have further gained connectivity. There are also idiosyncratic changes, such as the connectivity gain of Belgrade, which may both attributed to altered geopolitical circumstances which diminished its connectivity in 2010.

### 3.2. Connecting Cities in Asia and Europe

Table 4 gives an overview of the 20 strongest and the 20 weakest RCDC$_{a-i}$ connections. Although there is no explicit national, regional or geographical component in our specification of inter-city connections, there are nonetheless again strong geographical patterns in our findings: none of the weakest connections is within the same world region, all of the strongest connections are within the same world region. As indicated by Rugman and Verbeke [2005] and echoing the findings of Burger et al. [2013], there are few putatively 'global' services firms: the geographical scope of most of these firms is world regional rather than global, and there are therefore only a few truly 'global cities' [58,59]. The weakest connections are between second-tier European cities and Chinese cities. The strongest connections are Chinese cities that are strongly interconnected through Chinese banks. The largest of these Chinese firms have 'gone global' [43,60,61] but remain first and foremost 'national banks' that are distributed across the Chinese urban system due to the scale of Chinese market and population. As a result, many of the 'global' service connections of second-tier Chinese cities are in reality national flows. This is clearly an additional way in which states and larger regional schemes guide inter-city connectivity.

However, this overview of strongest and weakest connections also serves to show that if we want to examine the strongest connections between Asian and European cities, we must glean them from further down the distribution. Table 4 therefore lists the 20 strongest connections between Asian and European cities. With the exception of strong connections between Paris/London and Shanghai and the special case of the Special Administrative Region of Hong Kong, China is absent from this ranking. This suggests that although Chinese cities in general and leading Chinese world cities in particular have become much more connected overall, there is no marked orientation towards European cities: the connections are there, but they do not stand out in the connectivity profiles of European and Chinese cities despite the growing trade and investment between them. This is

above all relevant for understanding Beijing, which is totally absent from this ranking; its sizable global connectivity is to a large degree shaped by the headquarter functions of Chinese financial services firms, so its connectivity has a stronger national orientation than Shanghai's [62]. Instead, the strongest connections between Asia and Europe are for Singapore, Hong Kong, and Tokyo; they are involved in 14 out of 20 connections. Many of these connections are shaped by financial services firms, as evidenced by the presence of financial services firms (important for Frankfurt) and management consultancy firms (important for Düsseldorf). The dominance of Singapore, Hong Kong, Tokyo, and Singapore reaffirms the presence of a polycentric system in Asia, where there is not a single city, but rather a complementary range of cities that play a prominent role in connecting Asia and Europe.

**Table 4.** An overview of the 20 strongest and the 20 weakest RCDCa-i connections.

| | Strongest Links | | Weakest Links | | Strongest Inter-Regional Links | |
|---|---|---|---|---|---|---|
| 1 | Jinan | Xi'an | Changsha | Geneva | Paris | Singapore |
| 2 | Zhengzhou | Jinan | Nanjing | Geneva | London | Singapore |
| 3 | Changsha | Jinan | Brussels | Xi'an | Paris | Hong Kong |
| 4 | Jinan | Shenyang | Shenyang | Copenhagen | Hong Kong | London |
| 5 | Zhengzhou | Xi'an | Helsinki | Shenyang | Singapore | Frankfurt |
| 6 | Changsha | Xi'an | Helsinki | Chongqing | Frankfurt | Hong Kong |
| 7 | Jinan | Wuhan | Oslo | Xi'an | Tokyo | Paris |
| 8 | Jinan | Dalian | Zhengzhou | Madrid | Frankfurt | Tokyo |
| 9 | Changsha | Zhengzhou | Chongqing | Madrid | Shanghai | Paris |
| 10 | Xi'an | Wuhan | Xi'an | Helsinki | London | Shanghai |
| 11 | Shenyang | Xi'an | Chongqing | Geneva | Tokyo | London |
| 12 | Xiamen | Jinan | Xi'an | Geneva | Brussels | Singapore |
| 13 | Wuhan | Zhengzhou | Copenhagen | Jinan | Seoul | Paris |
| 14 | Jinan | Nanjing | Xi'an | Madrid | Düsseldorf | Singapore |
| 15 | Wuhan | Changsha | Belgrade | Dalian | Singapore | Moscow |
| 16 | Shenyang | Zhengzhou | Zagreb | Dalian | Bangkok | Paris |
| 17 | Jinan | Hangzhou | Changsha | Madrid | Paris | New Delhi |
| 18 | Shenyang | Changsha | Sofia | Dalian | Düsseldorf | Tokyo |
| 19 | Hangzhou | Xi'an | Jinan | Madrid | Brussels | Bangkok |
| 20 | Xi'an | Dalian | Shenyang | Madrid | Madrid | Singapore |

The marked importance of regionality in city connections also shows in our final set of results; Table 5 ranks all cities in our analyses based on their level of ROa. All European cities have stronger connections within Europe, and most Asian cities have stronger connections within Asia. In Europe, the least marked regional orientation towards other European cities can be found in cities with sizable international financial services complexes (London, Luxembourg, Frankfurt, and Dublin), at the eastern fringes of Europe (Moscow and St Petersburg), and major world cities more generally (London, Frankfurt, Paris, and Amsterdam). Cities such as Luxembourg and London are, at least in terms of their connectivity in corporate networks of service firms, almost as oriented towards as Asia as they are towards Europe. Meanwhile, the largest intra-European connections can above all be found in second-tier European cities, with—echoing some of the findings in Table 2—Madrid being very strongly oriented towards Europe. In Asia, we can discern three groups of cities. First, there is a group of Mainland Chinese cities that are strongly oriented towards Asia, above all China itself. This group includes Beijing for reasons outlined above. Second, there is a group of cities that show a balance between Asian and European connections. This includes Hong Kong, Singapore, and Shanghai, the only Mainland Chinese city that is not very 'Asian' or 'Chinese' in its business connections. Third and finally, there is a group of cities that are more strongly connected to Europe than to Asia. This includes all Indian cities (Delhi, Bangalore, Mumbai, Pune, and Chennai) alongside Dhaka and Almaty, as well as Bangkok, Manila, and Kuala Lumpur.

**Table 5.** Cities' ranks based on their level of ROa.

| Rank | City | World Region | Dominant Orientation | Dominant Orientation | Rank | City | World Region | Dominant Orientation | Dominant Orientation |
|---|---|---|---|---|---|---|---|---|---|
| 1 | Changsha | Asia | 1.99 | Asia | 16 | Zagreb | Europe | 1.37 | Europe |
| 2 | Shenyang | Asia | 1.96 | Asia | 17 | Madrid | Europe | 1.36 | Europe |
| 3 | Chongqing | Asia | 1.95 | Asia | 18 | Oslo | Europe | 1.35 | Europe |
| 4 | Xi'an | Asia | 1.92 | Asia | 19 | Copenhagen | Europe | 1.35 | Europe |
| 5 | Jinan | Asia | 1.90 | Asia | 20 | Kiev | Europe | 1.34 | Europe |
| 6 | Zhengzhou | Asia | 1.87 | Asia | 21 | Barcelona | Europe | 1.33 | Europe |
| 7 | Wuhan | Asia | 1.84 | Asia | 22 | Hamburg | Europe | 1.33 | Europe |
| 8 | Dalian | Asia | 1.83 | Asia | 24 | Bucharest | Europe | 1.33 | Europe |
| 9 | Hangzhou | Asia | 1.83 | Asia | 25 | Lisbon | Europe | 1.32 | Europe |
| 10 | Xiamen | Asia | 1.81 | Asia | 26 | Geneva | Europe | 1.32 | Europe |
| 11 | Nanjing | Asia | 1.77 | Asia | 27 | Sofia | Europe | 1.32 | Europe |
| 12 | Tianjin | Asia | 1.65 | Asia | 28 | Budapest | Europe | 1.32 | Europe |
| 13 | Chengdu | Asia | 1.58 | Asia | 29 | Munich | Europe | 1.31 | Europe |
| 14 | Shenzhen | Asia | 1.52 | Asia | 30 | Rome | Europe | 1.30 | Europe |
| 23 | Guangzhou | Asia | 1.33 | Asia | 31 | Zurich | Europe | 1.30 | Europe |
| 46 | Beijing | Asia | 1.22 | Asia | 32 | Stockholm | Europe | 1.29 | Europe |
| 55 | Osaka | Asia | 1.07 | Asia | 33 | Düsseldorf | Europe | 1.29 | Europe |
| 56 | Shanghai | Asia | 1.07 | Asia | 34 | Bratislava | Europe | 1.29 | Europe |
| 57 | Ho Chi Minh City | Asia | 1.05 | Asia | 35 | Warsaw | Europe | 1.29 | Europe |
| 58 | Hong Kong | Asia | 1.04 | Asia | 36 | Berlin | Europe | 1.28 | Europe |
| 60 | Jakarta | Asia | 0.99 | Europe | 37 | Belgrade | Europe | 1.27 | Europe |
| 61 | Hanoi | Asia | 0.98 | Europe | 38 | Vienna | Europe | 1.27 | Europe |
| 62 | Taipei | Asia | 0.98 | Europe | 39 | Brussels | Europe | 1.27 | Europe |
| 63 | Singapore | Asia | 0.97 | Europe | 40 | Prague | Europe | 1.26 | Europe |
| 64 | Karachi | Asia | 0.96 | Europe | 41 | Milan | Europe | 1.25 | Europe |
| 65 | Seoul | Asia | 0.96 | Europe | 42 | Lyon | Europe | 1.25 | Europe |
| 66 | Tokyo | Asia | 0.95 | Europe | 43 | Manchester | Europe | 1.23 | Europe |
| 67 | Kuala Lumpur | Asia | 0.93 | Europe | 44 | Stuttgart | Europe | 1.23 | Europe |
| 68 | Almaty | Asia | 0.91 | Europe | 45 | Nicosia | Europe | 1.22 | Europe |
| 69 | Pune | Asia | 0.91 | Europe | 47 | Paris | Europe | 1.20 | Europe |
| 70 | Manila | Asia | 0.90 | Europe | 48 | Amsterdam | Europe | 1.19 | Europe |
| 71 | Dhaka | Asia | 0.90 | Europe | 49 | Athens | Europe | 1.19 | Europe |
| 72 | Chennai | Asia | 0.88 | Europe | 50 | Dublin | Europe | 1.18 | Europe |
| 73 | Mumbai | Asia | 0.88 | Europe | 51 | Moscow | Europe | 1.17 | Europe |
| 74 | Bangalore | Asia | 0.86 | Europe | 52 | London | Europe | 1.11 | Europe |
| 75 | Bangkok | Asia | 0.86 | Europe | 53 | Frankfurt | Europe | 1.08 | Europe |
| 76 | New Delhi | Asia | 0.83 | Europe | 54 | St Petersburg | Europe | 1.07 | Europe |
|  |  |  |  |  | 59 | Luxembourg | Europe | 1.04 | Europe |

## 4. Discussion and Conclusions

What are the broader implications of our empirical findings? In any case, it is crucial to emphasize that our analytical framework presents only one possible approach to the analysis of city networks. For example, research on Asia–Europe connections in the context of the BRI [63–71], although sharing a similar metageographical outlook, focuses on different types of networks altogether. These and other city network approaches should be seen as complementary, but each have their strengths and limitations.

The first implication of our findings is that inter-city connections clearly are not a zero-sum game: the rising levels of connectivity of Asia's cities have not come at the expense of European cities. There is therefore clearly an opportunity for mutual strengthening, a discourse that interestingly also pervades the BRI (albeit with a geo-political twist). In our approach, it is firms rather than cities that are the actors of change and the key of of inter-city connections is cooperation rather than interurban competition for resources, capital, and knowledge. This does not imply that there is no intercity competition within their networking processes [39,72], but we argue that this cooperation process can be foregrounded because it involves the basic model of intercity connections: cities survive in networks and networks are reproduced through shared complementarities [73,74].

The second implication of our findings is that although our focus is firmly on intercity connections, we have shown that—to put it in Castells' [1996] terms—spaces of places (territorial mosaics, such as states or areal regions) inevitably interact with spaces of flows

(network configurations, such as intercity networks) [75]. In his research on Southeast Asia, Dick [2005] advanced the idea that any territorial framework should be approached as an open system, as interactions across national or regional borders become increasingly important in empirical and formative terms [30]. By using a city network perspective, we evaded state and regional forms of territoriality in our specification of Asia-Europe connections, but our results clearly show that no matter how powerful forces of globalization and city networks may have become, they will always be party to creating and reproducing more than just a global scale of activities.

The foremost scale is that of states, who remain major shapers of markets and creators of economies. All of our findings attest to this in some form; the presence of national gateways, the impact of the level of political centralization, the scale of national markets and urban systems, and idiosyncratic state histories are visible in city network patterns at the global scale. The reason why countries are key shapers of connectivity patterns is that states influence different producer services sectors in distinctive ways. For the various financial services there are regulations whose level of control varies by country. For example, in spite of China slowly allowing the internationalization of its official currency, the renminbi (RMB), controls on foreign financial institutions operating within China remain in place—this clearly shapes the networks of Chinese and non-Chinese financial services firms alike. For law, the state constitutes a legal jurisdiction that must be dealt with in transnational commercial projects [76]. States are also legitimate professional gatekeepers that determine who can and who cannot practice law, and other professions, in their territory. For management consultancy and advertising, the state is somewhat less invasive, but there are nonetheless national specifics. For examples, there are cultural differences in how products will be received. Global advertising has to deal with consumers who speak different languages and have very different responses to visual signals [77]. Finally, global management consultancy firms may need to overcome different business norms and cultures. Taken together, these examples show that even in global praxis of producer services, countries cannot be ignored.

However, the intersection of city networks and territorial configurations does not stop there. 'Asia' and 'Europe' are themselves metageographical areal constructions [78] that can be disassembled into a range of other, overlapping regional configurations. For example, in his discussion of Southeast Asia, Dick [2005] posits the idea of city networks intersecting with different regional framings [30]: "(f)lows of people, goods, money and information reveal that modern Southeast Asia is a network of cities, a subset of broader networks of cities that may be labelled as East Asian, Asian or Asia-Pacific". There is a substantive body of literature showing the continued manifestation of multi-layered regional patterns within 'global' networks of trade [79], investment [80], and transport and logistics [81]. Such a multilayered regionalism also emerges in recent research on the business connections of cities: Indraprahasta and Derudder [2017], for example, show that the geography of Jakarta's business connections is characterized by strong connections with major world cities alongside clear-cut Southeast Asian, East Asian, and national connections [33]. It can thus be said that our findings show both the enduring relevance of overlapping regional areal framings and the increasing relevance of what Thrift [1999] has called the 'blizzard of transactions' across the world [82].

A third and final implication of our findings is that, in spite of the continued importance of states and our focusing on business connections, this type of approach brings life to the more general idea of bringing more 'political space to cities' [83]. Major cities in China have often served as vanguards and engines of China's economic growth in an increasingly urbanized world [84]. In global urban studies, the novelty of urban processes emerging in Chinese cities is not easily covered by Western urban theory [85]. For example, with its top-down and centralized political system, the Chinese Central Government creates a perspective of competition among mayors by "evaluating" them on the basis of relative "economic success" [86]. This raises the option that the central government can reward city officials to focus more on both local and global specialization. The city mayors in China

could face pressure from the central government to take central roles in setting up specific transnational networks. The recently launched mayors' initiatives in cities' development plans further support such tendency: Beijing wishes to be the worlding leading harmonious and livable city, Shanghai a 'socialist modern international metropolis with leading world influence', Guangzhou a global benchmark city in digitalization, and Shenzhen a 'city of innovation and Entrepreneurship with global influence' [87–90].

After being so long under the domination of states, cities have yet to fully realize their newfound power. For example, new powerful mayors representing local government in alliance with network capital can produce a new city network governance regime. In his often-cited account of 'mayors ruling the world', Barber [2012] presented city mayors as the solution to what he sees as largely dysfunctional and paralyzed national governments [91]. Although this point of view is both provocative and meaningful, it is too simplistic [92]. Nonetheless, as explained by Acuto [2013], political science, and especially its international relations variant, has generally been too inattentive to the role of cities in global politics for the majority of the 1990s and 2000s or has rarely attributed agency to urban actors [93]. Only recently has the debate in political science circles started to delve into questions regarding forms of urban agency, but these are mostly framed as a matter of how mayors may marshal urban leadership in going beyond the stasis in existing global governance structures. The time is ripe for a more nuanced and contextualized answer to the question of how cities (can) act politically on the global scale in general and in Asia–Europe relations in particular.

**Author Contributions:** Conceptualization, B.D. and X.F.; methodology, B.D. and P.J.T.; validation, R.S.; formal analysis, B.D. and R.S.; resources, B.D. and X.F.; writing—original draft preparation, B.D. and X.F.; writing—review and editing, W.S.; funding acquisition, X.F. All authors have read and agreed to the published version of the manuscript.

**Funding:** This research was funded by the Ministerial Social Science Research Project of Ministry of Culture and Tourism of China, grant number 22DY17.

**Institutional Review Board Statement:** Not applicable.

**Informed Consent Statement:** Not applicable.

**Data Availability Statement:** Not applicable.

**Conflicts of Interest:** The authors declare no conflict of interest.

## Appendix A

**Table A1.** 175 service firms' networks.

| Law | Consultancy | Advertising | Accountancy | Finance |
|---|---|---|---|---|
| Kirkland & Ellis | McKinsey & Company | Accenture Interactive | Deloitte | ICBC |
| Latham & Watkins | Boston Consulting Group | PwC Digital Services | PwC | China Construction Bank |
| Baker McKenzie | Bain & Company | Deloitte Digital | EY | Agricultural Bank of China |
| DLA Piper | Deloitte Consulting LLP | IBM iX | KPMG | Bank of China |
| Skadden, Arps, Slate, Meagher & Flom LLP | Oliver Wyman | Cognizant Interactive | BDO | Wells Fargo |
| Dentons | Booz Allen Hamilton | BlueFocus(China) | RSM | Bank of America |
| Clifford Chance | EY-Parthenon | McCann Worldgroup | Grant Thornton | CITI |
| Sidley Austin LLP | Strategy& | Wunderman Thompson | Crowe | JP Morgan Chase |
| Linklaters | A.T. Kearney | Dentsu Aegis Network | Nexia International | China Merchants Bank |
| Allen & Overy | GE Healthcare | DDB Worldwide Communications Group | Baker Tilly International | HSBC |
| Morgan, Lewis & Bockius | Putnam Associates | Publicis Sapient | HLB | TD |

**Table A1.** *Cont.*

| Law | Consultancy | Advertising | Accountancy | Finance |
|---|---|---|---|---|
| Jones Day | Clearview Healthcare Partners | TBWA Worldwide | Kreston International | RBC |
| White & Case | KPMG LLP (Advisory) | Ogilvy | Mazars | Bank of Communication |
| Norton Rose Fulbright | The Bridgespan Group | Epsilon-Conversant | PKF International | Capital One |
| Freshfields Bruckhaus Deringer | Analysis Group | BBDO Worldwide | ETL Global | Shanghai Pudong Development Bank |
| Gibson, Dunn & Crutcher | LEK Consulting | Havas Creative Group | UHY International | Postal Savings Bank |
| Ropes & Gray | The Keystone Group | Publicis Worldwide | Russell Bedford International | BNP Paribas |
| CMS (EEG) | ghSMART | Omnicom Precision Marketing Group | Shinewing International | Sberbank |
| Greenberg Traurig | Insight Sourcing Group | Advantage Marketing Partners | Ecovis International | China CITIC Bank |
| Simpson Thacher & Barltlett | Alvarez & Marsal | Hakuhodo | Reanda International | SMBC |
| Weil, Gotshal & Manges | Gartner | Leo Burnett Worldwide | UC&CS America | Goldman Sachs |
| Paul, Weiss, Rifkind, Wharton & Garrison | Roland Berger | Tag | TGS Global | ING |
| Sullivan & Cromwell | Cornerstone Research | RRD Marketing Solutions | Parker Russell International | Barclays |
| Mayer Brown | Health Advances | FCB (Foote, Cone & Belding) | Auren | Industrial Bank |
| | | | | Scotiabank |
| | | | | China Everbright Bank |
| | | | | China Minsheng Bank |
| | | | | BMO |
| | | | | BBVA |
| | | | | MUFG |
| | | | | UBS |
| | | | | Morgan Stanley |
| | | | | U.S. Bank |
| | | | | DBS |
| | | | | Ping An Bank |
| | | | | CIBC |
| | | | | Rabobank |
| | | | | PNC |
| | | | | Société Générale |
| | | | | Commonwealth Bank of Australia |
| | | | | Merrill Lynch |
| | | | | Lloyds Bank |
| | | | | Crédit Suisse |
| | | | | Itaú |
| | | | | Mizuho Financial Group |
| | | | | Bradesco |
| | | | | Discover Bank |
| | | | | Intesa Sanpaolo |
| | | | | State Bank of India |
| | | | | QNB |
| | | | | NatWest-National Westminster Bank |
| | | | | HDFC Bank |
| | | | | Standard Chartered |
| | | | | Crédit Agricole |
| | | | | Crédit Mutuel |
| | | | | OCBC Bank |
| | | | | Caixa |
| | | | | NAB (National Australian Bank) |
| | | | | UOB (United Overseas Bank) |
| | | | | Nordea |
| | | | | Shinhan Financial Group |
| | | | | ANZ |
| | | | | Banco do Brasil |
| | | | | KBC |

**Table A1.** *Cont.*

| Law | Consultancy | Advertising | Accountancy | Finance |
|---|---|---|---|---|
| | | | | KB Financial Group |
| | | | | Emirates NBD |
| | | | | ABN AMRO |
| | | | | Hua Xia Bank |
| | | | | First Abu Dhabi Bank |
| | | | | Maybank |
| | | | | BNY Mellon |
| | | | | Westpac |
| | | | | Bank of Beijing |
| | | | | JP Bank |

## Appendix B

**Table A2.** 707 cities (European cities in bold and Asian cities in italics), rank-ordered by GNC$_a$.

| City | City | City | City | City |
|---|---|---|---|---|
| **London** | **Warsaw** | **Prague** | Denver | Panama City |
| New York | *Seoul* | **Lisbon** | Beirut | *Wuhan* |
| *Hong Kong* | Johannesburg | Miami | *Ho Chi Minh City* | **Geneva** |
| *Singapore* | **Zurich** | Dallas | **Athens** | **Manchester** |
| *Shanghai* | Melbourne | Washington DC | *Tianjin* | *Osaka* |
| *Beijing* | Istanbul | Houston | Abu Dhabi | Calgary |
| Dubai | *Bangkok* | **Hamburg** | Perth | **Stuttgart** |
| **Paris** | **Stockholm** | Bogota | Casablanca | **Belgrade** |
| *Tokyo* | **Vienna/Wien** | **Rome** | **Kiev** | Monterrey |
| Sydney | *Guangzhou* | **Berlin** | Montevideo | Kuwait City |
| Los Angeles | **Dublin** | *Chengdu* | **Helsinki** | *Changsha* |
| Toronto | San Francisco | **Barcelona** | **Oslo** | Tampa |
| *Mumbai/Bombay* | *Taipei* | **Düsseldorf** | *Chennai* | Caracas |
| **Amsterdam** | Buenos Aires | Tel Aviv | Philadelphia | **Sofia** |
| **Milan** | **Munich** | **Bucharest** | *Nanjing* | **Bratislava** |
| **Frankfurt** | **Luxembourg** | Doha | Seattle | Minneapolis |
| Chicago | Montreal | **Budapest** | *Hanoi* | San Jose |
| Sao Paulo | Boston (Massachusetts) | **Copenhagen** | Cape Town | **Zagreb** |
| *Kuala Lumpur* | *New Delhi* | Lima | *Hangzhou* | *Zhengzhou* |
| Mexico City | Santiago | Vancouver | Nairobi | *Dhaka/Jahangir Nagar* |
| **Madrid** | *Manila* | Brisbane | Manama | *Xiamen* |
| **Moscow** | *Shenzhen* | Atlanta | *Karachi* | *Shenyang* |
| *Jakarta* | *Bangalore* | Cairo | Rio De Janeiro | Tunis |
| **Brussels** | Riyadh | Auckland | *Chongqing* | *Almaty* |
| San Diego | Phoenix | Durban | Accra | Hartford |
| **Lyon** | **Antwerp** | **Vilnius** | Asuncion | Raleigh |
| **Nicosia** | **Rotterdam** | **Nantes** | Maputo | **Birmingham** |
| *Xi'an* | **Porto** | Ankara | Douala | **Krakow** |
| *Dalian* | Adelaide | San Juan | Nassau | Curitiba |
| Amman | **Baku** | **Wroclaw** | *Fuzhou* | **Seville** |
| **St Petersburg** | Guadalajara | Ottawa | Harare | Abuja |
| Guatemala City | *Qingdao* | Santo Domingo | **Poznan** | Tijuana |
| Lagos | **Ljubljana** | **Turin** | Kansas City | Port of Spain |
| Quito | **Belfast** | **Malmö** | Luanda | Abidjan |
| *Jinan* | **Cologne** | Dakar | Columbus | Belo Horizonte |
| Detroit | Algiers | **Bristol** | Milwaukee | *Ningbo* |
| San Jose | *Suzhou* | Nashville | **Katowice** | San Antonio |
| *Pune* | Medellin | **Tirana** | *Nagoya* | Brasilia |
| St Louis | *Islamabad* | **Valencia** | Sacramento | *Johor Bahru* |
| San Salvador | **Glasgow** | *Colombo* | Edmonton | *Yangon* |
| Kampala | *Phnom Penh* | *Taizhong* | **Málaga** | Puebla |

**Table A2.** *Cont.*

| City | City | City | City | City |
|---|---|---|---|---|
| *Calcutta* | *Kunming* | **Bilbao** | Queretaro | Cincinnati |
| *Hyderabad* | **Tbilisi** | Guayaquil | Salt Lake City | **The Hague** |
| Muscat | **Riga** | Managua | *Penang* | **Yerevan** |
| **Edinburgh** | Baltimore | La Paz | *Harbin* | **Strasbourg** |
| George Town | *Hefei* | Wellington | *Kaohsiung* | *Macao* |
| *Lahore* | *Ahmedabad* | Tegucigalpa | Indianapolis | Dammam |
| Jeddah | Dar Es Salaam | *Haikou* | **Lausanne** | **Leeds** |
| Austin | Orlando | Port Louis | **Limassol** | Lusaka |
| Charlotte | **Gothenburg** | Cleveland | *Taiyuan* | *Ulan Bator* |
| Porto Alegre | **Minsk** | **Montpellier** | Santa Cruz | Haifa |
| **Tallinn** | Aguascalientes | Tulsa | Mexicali | Palo Alto |
| *Cebu* | Christchurch | **Podgorica** | **Lille** | Baghdad |
| *Astana* | Jacksonville | Valencia | **Bordeaux** | **Cardiff** |
| **Bologna** | Richmond | **Lodz** | Bursa | Barranquilla |
| Portland | **Skopje** | Winnipeg | *Hsinchu City* | **Mannheim** |
| **Marseille** | Campinas | Buffalo | **Dresden** | Chihuahua |
| Canberra | Oklahoma City | **Graz** | Libreville | Memphis |
| **Naples** | **Toulouse** | Halifax | Quebec | Omaha |
| **Leipzig** | *Tashkent* | **Genoa** | Port Harcourt | **Bern** |
| Pittsburgh | Alexandria | Louisville | **Nice** | *Tainan* |
| **Utrecht** | *Zhuhai* | **Linz** | **Arhus** | Honolulu |
| **Newcastle** | Des Moines | *Fukuoka* | New Orleans | *Dushanbe* |
| **Nürnberg** | San Luis Potosí | Rochester | *Labuan* | Kabul |
| Mérida | **Chisinau** | Hamilton | **Bergen** | **Sheffield** |
| Ciudad Juarez | *Guiyang* | Windhoek | **Liege** | Kinshasa |
| *Surabaya* | Cordoba | *Vientiane* | **Basel** | Harrisburg |
| Cali | Leon | Recife | Jerusalem | Salvador |
| **Florence** | *Cochin/Kochi* | *Shijiazhuang* | *Hohhot* | **Kazan** |
| Las Vegas | *Changchun* | Pretoria | *Bandar Seri Begawan* | **Reykjavik** |
| Izmir | *Nanning* | Gaborone | Saskatoon | **Dortmund** |
| **Sarajevo** | Valparaíso | Port Elizabeth | *Lanzhou* | Goiania |
| *Urumqi* | *Nanchang* | Birmingham | **Bremen** | *Sapporo* |
| **Liverpool** | *Bishkek* | **Nottingham** | Rosario | Port Moresby |
| **Aberdeen** | **Southampton** | Kigali | Kingston | Hobart |
| **Hannover** | San Pedro Sula | *Wuxi* | **Grenoble** | *Kyoto* |
| *Novosibirsk* | *Yinchuan* | *Naha* | *Anshan* | Mombasa |
| Brazzaville | *Hamamatsu* | Cotonou | *Baotou* | **Rostov-on-Don** |
| **Essen** | Addis Ababa | *Gwangju* | **Bonn** | *Zhuzhou* |
| Blantyre | Mendoza | *Jilin* | *Luoyang* | *Handan* |
| *Kobe* | Torreón | *Taizhou* | *Takamatsu* | *Jaipur* |
| *Malacca* | *Vladivostok* | *Chittagong* | *Coimbatore* | *Huai'an* |
| *Yokohama* | Antananarivo | *Huizhou* | *Daejeon* | **Nizhny Novgorod** |
| Lomé | *Vadodara* | *Wuhu* | Peoria | Cartagena |
| **Palermo** | *Ufa* | Rabat | *Baoji* | Chattanooga |
| *Pusan* | *Wenzhou* | *Zhenjiang* | *Liuzhou* | N'Djamena |
| *Sendai* | Madison | Tucson | Anchorage | Vitoria |
| **Trieste** | *Tangshan* | *Medan* | *Daegu* | *Guilin* |
| Sanaa | Tripoli | Lilongwe | *Linyi* | *Mianyang* |
| Suva | *Nantong* | Bulawayo | *Davao* | Melbourne |
| Arbīl | **Leicester** | *Yantai* | *Zibo* | *Qinhuangdao* |
| *Shizuoka* | *Baoding* | **Plymouth** | *Nanyang* | Monrovia |
| *Xining* | Fortaleza | Mbabane | *Zunyi* | *Xingtai* |
| *Toyama* | Albuquerque | *Yancheng* | *Tai'an* | *Qingyuan* |
| *Chandigarh* | Maracaibo | Charleston | *Zhanjiang* | *Maoming* |
| **Norwich** | Malabo | Freetown | *Kumamoto* | Conakry |
| Norfolk | Victoria | *Xuzhou* | *Ma'anshan* | Kingston |
| Greensboro | **Duisburg** | *Rizhao* | *Hengyang* | Paramaribo |
| Providence | *Okayama* | *Xiangyang* | Lubumbashi | Barquisimeto |

**Table A2.** *Cont.*

| City | City | City | City | City |
| --- | --- | --- | --- | --- |
| *Kathmandu* | *Yangzhou* | *Changzhou* | *Daqing* | **Mainz** |
| *Yekaterinburg* | *Weifang* | *Shantou* | *Lianyungang* | *Datong* |
| *Hiroshima* | *Yichang* | *Putian* | *Bandung* | *Zhangjiakou* |
| *Bengbu* | *Kaifeng* | **Voronež** | *Batam* | *Patna* |
| *Multan* | *Nanchong* | Stockton | **Heidelberg** | *Visākhapatnam* |
| Yaonde | **Swindon** | Cuernavaca | Adana | *Gujranwala* |
| McAllen | Georgetown | Allentown | Kayseri | *Quetta* |
| *Semarang* | **Karlsruhe** | Antalya | *Trivandrum* | *Chelyabinsk* |
| Niamey | *Changshu* | Danbury | **Dnipropėtrovs'k** | Battle Creek |
| Oran | *Ashkhabad* | Bentonville | *Mysore* | **Samara** |
| *Xiangtan* | Ouagadougou | Bucaramanga | *Ulsan* | **Volgograd** |
| Winston-Salem | Djibouti | Maracay | *Makassar* | Havana |
| *Yiwu* | Kano | **Sandviken** | *Rayong* | *Mandalay* |
| Bamako | Cochabamba | *Wanzhou* | *Dehradun* | Port-Au-Prince |
| Manaus | *Huaibei* | **Clermont-Ferrand** | *Aurangābād* | Sūsah |
| *Jinzhou* | *Sūrat* | *Nāgpur* | Marrakesh | Mogadishu |
| *Jiaozuo* | Faisalabad | *Lucknow* | *Rājkot* | Kumasi |
| *Chifeng* | *Incheon* | *Nāshik* | *Denpasar* | Maseru |
| *Huainan* | *Taoyuan* | *Peshawar* | *Chiba* | Konya |
| *Benxi* | Kaduna | *Jodhpur* | Agadir | João Pessoa |
| *Xinxiang* | Toluca | **Odėsa** | *Jiangyin* | *Indore* |
| *Qiqihar* | Damascus | Khartoum | **Ludwigshafen** | Asmara |
| *Perm* | Ibadan | Belem | *Hyderabad* | Isfahan |
| *Kitakyushu* | El Paso | Maceió | *Yogyakarta* | **Wolfsburg** |
| *Krasnoyarsk* | Tehran | *Chonburi* | **Venice** | *Amritsar* |
| Little Rock | Virginia Beach | Gaza | *Kawasaki* | *Vijayawāda* |
| Acapulco | Gaziantep | *Jamshedpur* | *Guwāhāti* | *Sakai* |
| *Changwon* | Nouakchott | Teresina | *Ludhiāna* | *Cixi* |
| Benin City | *Tiruppūr* | Ḥalab | *Jalandhar* | *Khulna* |
| Maiduguri | *Vārānasi* | Aba | *Kānpur* | *Surakarta* |
| *Akita* | Fès | Ahvāz | *Kotā* | Santos |
| *Āgra* | **Leverkusen** | Al-Baṣrah | *Madurai* | Palembang |
| *Alīgarh* | **Pombal** | Al-Madīnah | *Meerut* | *Dili* |
| *Allahābād* | **Ruhrgebiet** | Al-Mawṣil | *Morādābād* | *Angeles* |
| *Asansol* | *Malang* | Bissau | *Ranchi* | *Bandar Lampung* |
| *Bareilly* | *Pekanbaru* | *Cangnan* | *Salem* | *Zamboanga* |
| *Bhilai* | Bangui | **Donetsk** | *Solāpur* | *Omsk* |
| *Bhopāl* | Natal | Kirkūk | *Srīnagar* | **Saratov** |
| *Bhubaneswar* | São Luís | Makkah | *Tiruchirāppalli* | *Surgut* |
| *Dhanbād* | *Rawalpindi* | Mashhad | Qom | Shīrāz |
| *Gwalior* | **Kharkov** | Mbuji-Mayi | *Serang* | Tabrīz |
| *Hubli-Dhārwār* | *Fuji* | Onitsha | *Pyongyang* | *Thimphu* |
| *Jabalpur* | Porto Novo | | | |

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
