# Peer review of "Connections between Asian and European World Cities: Measurement, Analysis, and Evaluation"

_land, doi:10.3390/land11091574_

Round 1
Reviewer 1 Report (Previous Reviewer 3)
Thank you for the corrections made. The introduced changes are in line with the guidelines I made recently and, in my opinion, filled the gaps in the previous text. Comments on this version of the text should be easy to introduce. There is a personal form in verse 199, the sentence should be rectified to impersonal. Table 1 has moved in relation to the header in the printout (but you will deal with it at the stage of typesetting). In Appendix it would be worth introducing a slightly different font, but it is also a technical matter. In other words, after slight corrections, the text, in my opinion, can be printed.
Author Response
Please see the attachment.

Reviewer 2 Report (Previous Reviewer 2)
The authors properly revised the manuscript according to with my previous comments
Author Response
Thank you for this generous and supportive appraisal.
This manuscript is a resubmission of an earlier submission. The following is a list of the peer review reports and author responses from that submission.
Round 1
Reviewer 1 Report
Dear authors,
I hope you're well. I appreciate your scientific effort to produce this manuscript and have some comments to help you improve your paper:
The manuscript ´Connections between Asian and European world cities: measurement, analysis, and evaluation´ contains the research results in applying the model conjecturing inter-city connections and analyzing the networks of major cities in Asian and European regions. The results confirm the relevance of the proposed methodology as an instrument for land management.
The manuscript’s strengths:
The general approach of the manuscript is good. The manuscript is informative and good structured. The title matches the content. The topic fits the Land journal scope and the case is relevant. The introduction and literature review provide sufficient background and include sufficient references. The analysis has been performed reliably. The conclusions match the research idea. Overall, the work deserves an average rating.
The manuscript’s weaknesses:
Abstract. There is a lack of research background (research problem/gap). I suggest adding this aspect to make readers know why your research is necessary to be conducted. This clarification can make your research objective clearer.
Introduction. More than 50% of the cited references are above the last 5 years. It is suggested to improve the reference list through using mostly resent publications. Besides, the introduction doesn´t include a research hypothesis, it should be better to add it.
Methodology. The details, given in the methods section are too low for the results reproducibility. It is recommended to add the research scheme and expand the equations: Equation (1) - please add the legend.
Results. There are missing data in the Results section. Page 7: There is no Table 2, only the table reference (please see the last paragraph), Subsection 3.2: there is only the references to Table 3 and Table 4.
Discussion. The section is too brief and suggested to expand it further. Authors should point out the marginal contribution of this study, the similarities and differences between this study and similar ones and the reasons for the differences, the policies and duties of this study on other regions, and the direction of further research in the future.
Reviewer 2 Report
Dear all,
The topic of this research is quite interesting; however, the manuscript and the research profoundness seems shortly for a n indexed thematic journal as LAND.
Therefore, in my opinion, the authors should expand the research in order to obtain more relevant outcomes. Besides, the used methods should be expalianed in a more detailed way (consider to use a methodologycal scheme); also, the literature review section should be expanded as well as the study limitations and future research lines should be emphatized.
Reviewer 3 Report
The subject matter is interesting, in my opinion, the whole thing requires far-reaching corrections and additions.
First (but not the most important) style - the whole thing should be written impersonally, that the authors did something the same way, or something they think should be described but in a nonpersonal way.
Secondly - the results are missing in the text (literally) the references to table 1 are misleading (because we only have measures in it), the remaining table (up to 4) is missing - or it has not been included in my version of the article. The effect is that we do not know which cities the study concerned (perhaps a map with their locations would be useful, since there are 707 of them), we do not know their location in relation to the infrastructure (existing and planned) within the BRI - and I think that one of the most important variables is because railways, roads and ports are at the core of this initiative.
Thirdly - we do not know anything about the criteria for selecting companies for the study, and this is a key assumption for the presented research, what industries were taken into account, what capital prevails in them, what is their size - the results of the research depend on it, the very statement that they are then, for example, Chinese banks say nothing
Fourth - There are repetitions in the text (why mention many times that we have an observation matrix with dimensions of 707 cities x 175 companies - in the last two paragraphs of point 2.1. We have it repeated 4 times). There are also errors in references to the literature in the text (e.g. reference 34 should probably be corrected to 30). We also have a confused interpretation of some of the results with the discussion - perhaps it would be worthwhile to combine the discussion elements from the final part with the discussion links in the previous part 3 and separate them into a relevant section devoted only to these issues.
Fifthly - the journal has a geographical focus and in this text there are very few references to areas. In addition to the aforementioned maps with the locations of the studied cities against the background of BRI infrastructure, a layer with distinguished ranges (e.g. where the Ural Mountains area ends and begins) as well as a map / maps with example locations of companies from a given industry would be useful.
However, pending the amendments, I would submit the article to be rejected, with the possibility of resubmitting it in a modified form.